# Scoping review of qualitative studies investigating reproductive health knowledge, attitudes, and practices among men and women across Rwanda

Julie M. Buser[1]*, Ella August[2], Gurpreet K. Rana[3], Rachel Gray[4], Olive Tengera[5], Faelan E. Jacobson-Davies[6], Madeleine Mukeshimana[7], Diomede Ntasumbumuyange[8], Gerard Kaberuka[9], Marie Laetitia Ishimwe Bazakare[10], Tamrat Endale[11], Yolanda R. Smith[12]

1 Center for International Reproductive Health Training (CIRHT), University of Michigan, Ann Arbor, Michigan, United States of America, 2 Department of Epidemiology, University of Michigan School of Public Health, PREPSS (Pre-Publication Support Service), Ann Arbor, Michigan, United States of America, 3 Global Health Coordinator, Taubman Health Sciences Library, University of Michigan, Ann Arbor, Michigan, United States of America, 4 Center for International Reproductive Health Training (CIRHT), University of Michigan, Ann Arbor, Michigan, United States of America, 5 Head of Midwifery Department, College of Medicine and Health Sciences, University of Rwanda, Kigali, Rwanda, 6 Center for International Reproductive Health Training (CIRHT), Department of Obstetrics and Gynecology, University of Michigan, Ann Arbor, Michigan, United States of America, 7 College of Medicine and Health Sciences, University of Rwanda, Kigali, Rwanda, 8 Department of Obstetrics & Gynecology, College of Medicine and Health Science, School of Medicine and Pharmacy, University of Rwanda, Kigali, Rwanda, 9 Research Coordinator, University of Rwanda, Kigali, Rwanda, 10 M&E Specialist, University of Rwanda, Kigali, Rwanda, 11 Center for International Reproductive Health Training (CIRHT), University of Michigan, Ann Arbor, Michigan, United States of America, 12 Center for International Reproductive Health Training (CIRHT), Department of Obstetrics and Gynecology, University of Michigan, Ann Arbor, Michigan, United States of America

* jbuser@umich.edu

**Data Availability Statement:** All relevant data are within the manuscript and its Supporting information files.

## Abstract

### Background

Research efforts in Rwanda to improve sexual and reproductive health and rights (SRHR) are increasing; however, comprehensive literature reviews on SRHR are limited. This scoping review examines individual and contextual factors shaping knowledge, attitudes, and practices in the domains of: 1) family planning, 2) abortion care, and 3) other SRHR in Rwanda. Recognizing that individual, community, and societal factors influence RH, this review is guided by Bronfenbrenner's *Ecological Systems Theory*.

### Methods

Eligible studies were conducted in Rwanda, included males and/or females of any age, and were published within the past 20 years. Studies reporting views of only healthcare or other professionals were excluded.

### Results

Thirty-six studies were included. The majority addressed individual and contextual considerations. At the individual level, studies explored knowledge about SRHR problems while at

**Funding:** This study was financially supported by Center for International Reproductive Health Training, University of Michigan in the form of a grant awarded to TE. No additional external funding was received for this study. The funder had no role in study design, data collection and analysis, decision to publish, or preparation of the manuscript.

**Competing interests:** The authors have declared that no competing interests exist.

**Abbreviations:** CAC, Comprehensive abortion care; EST, Ecological Systems Theory; FDGs, Focus group discussions; FP, Family planning; SRH, Sexual and reproductive health; SRHR, Sexual and reproductive health and rights.

the interpersonal level, the support and attitudes of men and community members for adolescent SRHR were investigated. In terms of healthcare organization, maternal health practices, increased access to family planning programs, and the need for sexually transmitted infection programs was explored. At the social and cultural level, researchers investigated beliefs and traditional gender roles. Regarding public health policy, studies mentioned promoting and increasing funding for SRHR and reducing gender inequities.

## Conclusion

Our findings can inform SRHR research programs, public health campaigns, and policy advances in Rwanda.

## Introduction

A synthesis of the qualitative evidence regarding the reproductive health of Rwandans is critical when considering research investments, school programs, and policy. Rwanda is Africa's most densely populated country with a high fertility rate of 4.2 children per woman and a high maternal mortality of 210/100,000 births [1]. Globally, family planning (FP) reduces maternal mortality by decreasing both total and higher-risk pregnancies [2]. Rwanda has made measurable success in bringing FP information and accessibility to its citizens [3] and is striving to achieve the highest attainable standard of health for everyone [4]. Rwanda is committed to achieving the Sustainable Development Goals by 2035 and declared FP and adolescent sexual and reproductive health (SRH) a national priority [5].

To achieve these goals, it is important to investigate factors shaping SRH in Rwanda. For the purpose of this scoping review, we operationalized SRH to include a state of complete physical, mental, and social well-being relating to the reproductive system, including sexuality education, FP, preconception care, antenatal and safe delivery care, postnatal care, STI/HIV prevention services, and preventive screening, early diagnosis and treatment of RH illnesses and cancers [6]. Improving SRH is crucial to eliminating health disparities, reducing rates of infectious diseases and infertility, and increasing educational attainment, career opportunities, and financial stability [7]. A clear understanding of SRH factors within one's own context is critical to advancing society.

While research efforts to understand knowledge, attitudes, and practice of SRH among Rwandans are increasing, comprehensive scientific literature reviews on these topics are limited and there remains a gap in knowledge. The aim of this qualitative scoping review is to examine individual and contextual factors shaping knowledge, attitudes, and practices in the domains of: 1) family planning, 2) abortion care, and 3) other sexual and reproductive health and rights in Rwanda. Recognizing that individual, community, and societal factors influence SRH, this review is guided by Bronfenbrenner's *Ecological Systems Theory* (EST) [8–11]. Within the EST, an individual is considered an integrated system in which psychological processes—cognitive, affective, emotional, motivational, and social—operate in coordinated interaction with one another [12]. We captured qualitative data to characterize how an individual's knowledge, attitudes, and practices reflect individual, community, and societal orientations related to SRH. The results from this qualitative scoping review can be used to inform future reproductive health research programs and the development of evidence-based guidelines and policy in Rwanda.

## Materials and methods

Our methodology was based on stages of the scoping review framework adopted by Arksey and O'Malley [13] to: (1) identify the research question; (2) identify relevant studies; (3) select studies; (4) chart the data; and (5) collate, summarize, and report the results. To assess study quality of the included literature, we followed the Critical Appraisal Skill Programme (CASP) checklists [14].

### Search strategy

A search of the literature was conducted by a health sciences informationist (GKR) in March 2022. The eight databases searched were MEDLINE (via Ovid interface), EMBASE (via Embase.com), Scopus, CINAHL (via EBSCOhost), Web of Science Core Collection (via Thomson Reuters), Global Health (via CABI), PsycINFO (via EBSCOhost) and Women's Studies International (via EBSCOhost).

Final search strategies were determined through test searching and use of search syntax to enhance search retrieval. The search strategies implemented in all eight databases can be accessed at https://hdl.handle.net/2027.42/175847. In seven of the databases, the search was limited to articles published from 2002 to 2022. In the Global Health database, due to the available search parameters in the CABI interface, the search strategy was limited to articles published from 2001 to 2021. We limited the timeframe to the past twenty years to identify themes in the current research literature while allowing time for the reinitiation of research activities after the Rwandan genocide.

A total of 948 citations were exported to EndNote for processing and removal of duplicate citations. There were 379 unique citations exported to Rayyan [17] for assessment and initial screening. Rayyan is a free web-based tool designed for researchers working on evidence synthesis projects, including scoping reviews and systematic reviews.

### Study selection

Studies were eligible for inclusion if they were original research publications conducted in Rwanda with Rwandan male and female citizens of any age. We only included studies with data from multiple countries if Rwanda-specific data were reported separately. Articles in any language that were conducted within the last 20 years were included. Any study designs that used qualitative data in peer-reviewed publications were included. Studies reporting views limited to only health care or other professionals were excluded as were those using only quantitative data; other publication types (dissertation, abstract, commentary, editorial, protocol papers); other focus areas (not in the domains of FP/CAC/SRHR); other study duration (published more than 20 years ago); and those conducted outside Rwanda. Articles were also identified via hand searching of reference lists of included studies (conducted by JMB, EA, OT, FGD, RG, GK, LI, TE, and YS).

### Screening process

Using Rayyan, title and abstract screening was performed by two independent researchers who were blinded to one another's assessments (JMB, EA, FJD, RG, TE, and YS). A third reviewer resolved discrepancies (JMB, EA, FJD, RG, TE, and YS). A similar process was used to review the full-text articles (JMB, EA, OT, FGD, RG, GK, LI, TE, and YS). Hand searching of reference lists was performed (JMB, EA, OT, FGD, RG, GK, LI, TE, and YS). The reviewing team communicated throughout the abstract and full-text screening processes to address any questions about study selection.

## Data collection

A data extraction form was used to collect author names, article title and year, participant characteristics (gender, age, etc.), study aims, sample size, findings, and implications (JMB, OT, FJD, RG, GK, LI). Researchers also used the data extraction form to identify the study domain (i.e., FP, SRHR, CAC) and EST level (i.e., individual, interpersonal, healthcare organization, social & cultural, and/or public health policy).

## Results

Fig 1 displays the PRISMA flow diagram, generated using a web-based tool created by Haddaway et al. [18]. A total of 36 studies were included in our final sample. Three of these articles were found through hand searching reference lists. S1 Appendix includes the citation counts for search strategies implemented on 3/3/2022. S2 Appendix shows the completed data extraction form filled in by our study authors. Regarding data collection type, in 15 studies, researchers conducted individual interviews while focus group discussions (FGDs) were conducted in 11 studies. Study domains of the articles in our sample are displayed in Fig 2 with SRHR the topic most frequently studied. Most studies included a sample size of less than 50 participants (Fig 2). The largest number of studies in our sample focused on the individual EST level (Fig 2) but often included discussion of multiple levels.

### Summary of sample study aims

In terms of study aims, several articles focused on the effectiveness and effects of different interventions: use of standard days method of FP based on a woman's menstrual cycle [19],

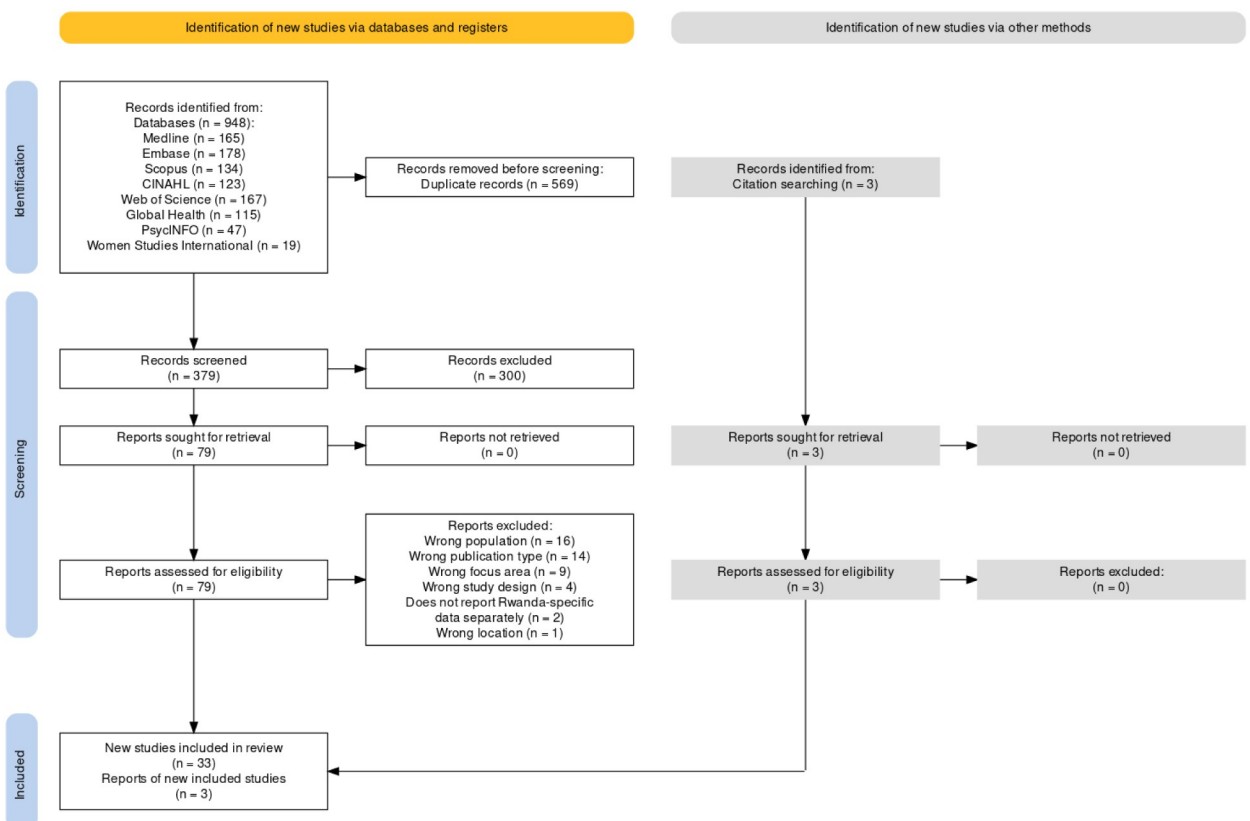

**Fig 1. PRISMA flow diagram of articles screened for scoping review of qualitative studies of reproductive health across Rwanda, 2001–2021.**

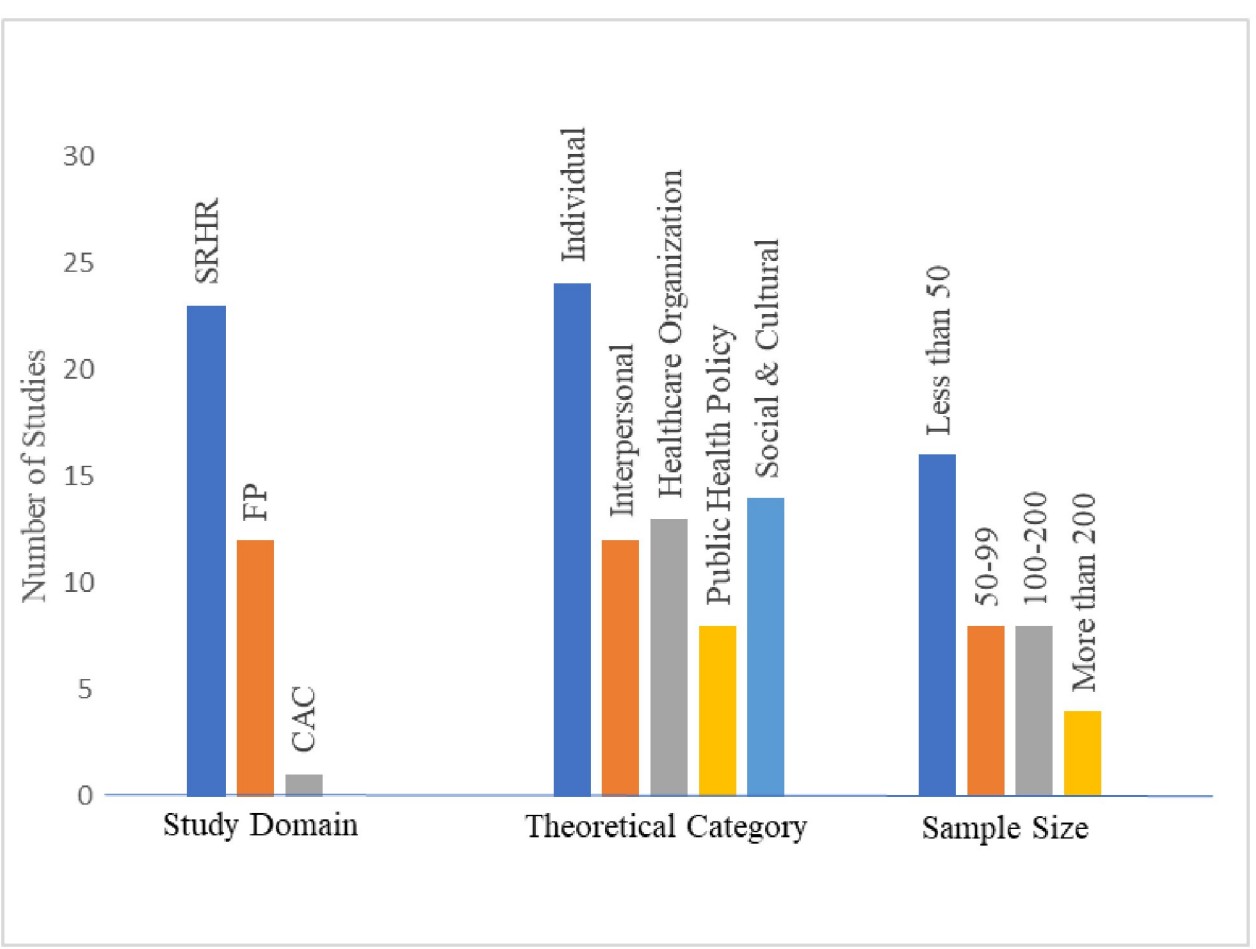

**Fig 2. Ecological systems theory study categories.**

engaging fathers in caregiving [20], implementation of coordinated public planning service delivery [21], group antenatal care [22], and an entertainment-education serial radio drama [23]. Three studies focused on understanding adolescents' perspectives or experiences of SRH social norms, specifically with: sexual coercion [24], adolescent pregnancy [25], and sexuality and relationships [26]. Additionally, Coast et. al. [27] evaluated the relationship between adolescents' age and their perceptions and experiences with SRH. While Doyle et al. [20], Tuyisenge et al. [28], Påfs, Musafili, et al. [29], and Ndirima et al. [30] focused on women's perceptions of the quality or accessibility of care; Stern & Heise [31] and Mumporeze et al. [32] investigated sexual power dynamics of "sexploitation" and sexual coercion, respectively.

Six studies focused on specific barriers to: modern contraception use [33], birth preparedness [34], antenatal care [35], positive client-provider relationships [36], collaborative couple contraception use [37], and on access and provision of maternal health services [28]. Tolley et al. [38] and Kestelyn et al. [39] explored the acceptability of different contraception types (long term injectable and vaginal ring, respectively) to sexually active HIV negative women between 18 and 35 years old. Adedimeji et. al. [40], Coast et al. [27], Doyle et al. [41] and Stern et al. [42] focused specifically on the effect of social and gender norms on individuals' health.

## Summary of included study implications framed by the ecological systems theory

The included studies addressed various implications in the domains of FP/CAC/SRHR in Rwanda that can be delineated within socio-ecological levels.

**Individual (Microsystem) and interpersonal (Mesosystem).**   Four articles discussed implications related to the role men play in maternal health [20, 37, 43, 44, 51]. These themes included the need to include men in discussions of FP and contraception and keeping them engaged in pregnancy discussions. Doyle and colleagues [20], maintain that gender-transformative programs, which engage men in deliberate questioning of gender norms, can increase men's involvement in ways that shift the burden of care work and address unequal power relations. Schwandt et al. [37] suggest that further increasing male involvement in contraceptive use via gender transformative approaches would be beneficial for Rwandan FP providers, men, and women. Meanwhile, in a study by Påfs, men saw an engaged and caring partner as the ideal yet faced challenges due to normative gendered expectations in Rwandan society [43].

To reduce barriers to modern contraceptive use for women in Rwanda, Brunie et al. [33] recommended increasing efforts to convey information aimed at men, developing effective messages about postpartum risk of pregnancy and training providers on postpartum contraceptive eligibility and needs, and reinforcing use of alternative pregnancy-screening methods. Mothers in the study by Musabyimana et al. [22] reported that group antenatal and postnatal care provides compelling benefits to women and families, but highlighted that additional resources (intensive reminder communications, human resources at the health center, and large-scale community outreach) must be factored into any future decision to scale a group care model. Ingabire et al. [45] highlighted the need to focus programs assisting women to leave sex work towards vulnerable house-girls and the need to include financial safety nets so that a time of financial difficulty does not necessitate a return to sex work.

**Healthcare organization (Exosystem).**   Seven articles articulated the need to increase access to and use of contraception and FP programs [36, 38, 39, 46–49]. Kestelyn et al. [39] and Farmer et al. [46] discussed the relationship between broader acceptance of FP by the society and the adoption of the methods by the population. Researchers maintain that as Rwanda continues to refine its FP policies and programs, it will be critical to address community perceptions around fertility and desired family size, health worker shortages, and stock-outs, as well as to engage men and boys, improve training and mentorship of health workers to provide quality services, and clarify and enforce national policies about payment for services at the local level [46]. Schwandt et al. [47] and Tolley et al. [38] discussed the role providers can play in disseminating accurate information about FP methods which will correct misconceptions and increase their use. Veldhuijzen et al. [49] found that condom use is low among Rwandan men and women and that condoms are mainly used by men during commercial sex, however, after discussing the use of microbicides, the study participants gained a better understanding of this contraception type and were more likely to use it.

Seven articles centered on the need for Rwanda's health system to support maternal health through providers ensuring that women feel respected, safe and able to advocate for themselves [29, 30, 34, 50] and in terms of policies, to be sure there are ways for women to receive the health care they need [43, 44, 51]. Ndirima and colleagues [30] imply that women's expectations, suggestions, and needs can enhance providers' awareness of the women's priorities during childbirth and serve as a guidepost for health services to increase the quality, acceptability and uptake of maternal health services. Kalisa et al. [34] suggest there is a need for addressing inconsistent health policies hindering the intention to access professional care in Rwanda.

According to Tuyisenge [43], interventions involving men are encouraged to increase their understanding of the implications of their involvement in maternal health without compromising women's autonomy in decision-making and to promote positive maternal health outcomes.

Four articles discussed adolescent SRH [24–27]. Coast et al. [27] and Michielsen et al. [26] spoke about young people being particularly at risk for adverse SRH outcomes, including HIV/STI infection or unwanted pregnancies, due to the societal context in which they engage in sexual relations. Van Decraen et al. [24] and Coast et al. [25] discussed the need to improve education and access to the health sector to protect adolescents from adverse outcomes associated with sexual coercion and teen pregnancy. Researchers maintain that concerted and multisectoral efforts—across education, justice and health—are critical to prevent sexual violence and unwanted adolescent pregnancy [25]. It is important to support adolescent mothers in hopes of reducing the likelihood that adolescent mothers and their children are left behind while preventing the intergenerational transmission of disadvantage [25].

**Social and cultural (Macrosystem) and public health policy (Chronosystem).**   Six articles described gender roles and possible ways to decrease gender inequalities by linking policy and societal norms [31, 32, 41, 42, 45, 52]. Mumporeze et al. [32] and Kubai & Ahlberg [52] discussed that Rwanda's legal system does not adequately protect women, despite the current gender equity policies. Policy shifts must be paralleled by shifts in the cultural expectation of gender roles. Doyle [41] suggests there is a need for capacity building efforts to challenge health providers' inequitable gender attitudes and practices and equip them to be aware of gender and power dynamics between themselves and their clients.

Two articles outlined Rwandan society's view of fertility [23, 53]. Shelus et al. [23] specifically discussed the need to convey fertility messages clearly and frequently, focusing on those central to the desired behavior change and monitor the frequency of these messages. As accurate information about fertility may contradict widespread cultural beliefs, listeners may not initially be receptive to the new information, such as postpartum pregnancy risk. Dohnt et al. [53] found that couples without children contribute to society by acting as caregivers, but infertility causes severe suffering and there is an urgent need to recognize infertility as a serious reproductive health problem and to put infertility care on the public health agenda.

Two articles evaluated sexually transmitted infections and ways to decrease their appearance in society [40, 54]. Adedimeji et al. [40] discussed the need for interventions to increase access to care that decrease both individual and societal determinants of risks and access to health services to limit the consequence of men having sex with men as a bridge for HIV transmission to the general population. An environment of intense social stigma and social isolation in Rwanda makes it difficult to obtain information or services to improve sexual health [40]. Veldhuijzen et al. [54] also discussed the benefit of STI programs that target individuals showing symptoms. They articulated that the reach of an STI control program could be increased by removing barriers to accessibility of care and increasing utilization of care by symptomatic patients.

## Synthesis of included studies

By framing implications of studies included in the review using the EST, we have situated their findings at various socio-ecological levels: individual attitudes, interpersonal relationships, healthcare organization, cultural and social factors, and public policy. Fig 3 shows the EST model inspired by Berger [55], Buser et. al. [56] and Stanger [57], adapted for this qualitative scoping review, incorporating the factors affecting RH in Rwanda and their relevance. The

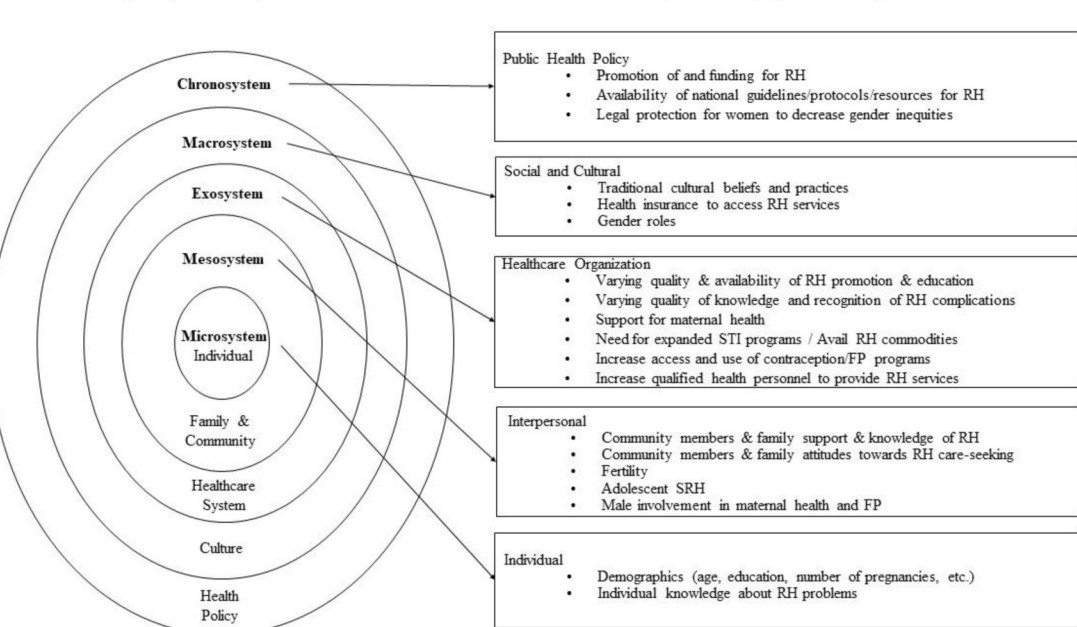

**Fig 3. Our adaptation of the ecological systems theory to characterize reproductive health in Rwanda.**

majority of studies in our sample addressed individual and contextual considerations at several EST levels.

## Discussion

### Summary of major findings

We conducted this qualitative scoping review to examine individual and contextual factors shaping the knowledge, attitudes, and practices in FP/CAC/SRHR in Rwanda. Included studies offered important implications at various EST levels. At the individual level, studies explored individual knowledge about SRH problems. Meanwhile, at the interpersonal level, male and community support and attitudes of SRH were investigated along with adolescent SRH. In terms of healthcare organization, maternal health practices, increased access to FP programs, and the need for expanded STI programs were explored in several studies. At the social and cultural level, researchers investigated cultural beliefs and traditional gender roles relating to SRH. Regarding public health policy, studies advocated promoting and increasing funding for SRH resources and affording better legal protection for women to reduce gender inequities.

Similar to findings in our review, researchers performing a scoping review on sexual and reproductive health behaviors among Tanzanian adolescents found the available published information shows that adolescents engage in high-risk sexual behaviors and experience adverse consequences [58]. Consistent with our review of studies in Rwanda, a mixed method scoping review by Wulifan and colleagues [59] demonstrated a noticeable gap regarding awareness and uptake of contraception leading to high unmet need for FP across LMICs. Also in line with our findings, a systematic review by Ivanova et al. [60] exploring knowledge, experiences, and access to SRH services in adolescent girls and young women across the African continent affected by conflict and disaster showed this population group

often experiences gender-based and sexual violence and abuse with limited access and availability of SRH services.

## Strengths and limitations of this qualitative scoping review

To our knowledge, this is the first qualitative scoping review of SRH knowledge, attitudes, and practice in Rwanda. A strength of this review is the high number of databases searched allowing us to provide an overview of the qualitative scientific literature pertaining to SRH in Rwanda. Another strength is that diverse types of settings were captured within Rwanda, including rural and urban areas, as well as coverage of men and women of all ages. While several databases were searched, with an ever-expanding knowledge environment, studies indexed in diverse sources, and unpublished data, it is possible that all relevant studies were not identified. Another potential limitation is that findings might not be generalizable to other areas in sub-Saharan Africa given the specific focus on Rwanda.

## Conclusions

This qualitative scoping review of SRH in Rwanda provided an overview of the published scientific literature. The included studies provided implications and important next steps for future research. In terms of policy changes, for Rwanda to achieve the Sustainable Development Goals by 2035, it is important to advocate at the national level for the prevention and treatment of violence against women, improved access to FP/SRH services, and expanded STI programs. Additional public health policy changes are needed to provide affordable legal protection to reduce gender inequities. We hope that the summarized findings of this scoping review will be used to inform future SRH research programs, public health campaigns, and policy advances in Rwanda.

## Supporting information

**S1 Appendix. Search citation counts.**
(DOCX)

**S2 Appendix. Data extraction form.**
(XLSX)

## Author Contributions

**Conceptualization:** Julie M. Buser.

**Data curation:** Julie M. Buser, Gurpreet K. Rana, Rachel Gray.

**Formal analysis:** Julie M. Buser, Ella August, Gurpreet K. Rana, Olive Tengera, Faelan E. Jacobson-Davies, Gerard Kaberuka, Marie Laetitia Ishimwe Bazakare, Tamrat Endale, Yolanda R. Smith.

**Investigation:** Julie M. Buser.

**Methodology:** Julie M. Buser.

**Project administration:** Rachel Gray, Tamrat Endale, Yolanda R. Smith.

**Software:** Faelan E. Jacobson-Davies.

**Supervision:** Julie M. Buser.

**Visualization:** Julie M. Buser, Faelan E. Jacobson-Davies.

**Writing – original draft:** Julie M. Buser.

**Writing – review & editing:** Ella August, Gurpreet K. Rana, Rachel Gray, Olive Tengera, Faelan E. Jacobson-Davies, Madeleine Mukeshimana, Diomede Ntasumbumuyange, Gerard Kaberuka, Marie Laetitia Ishimwe Bazakare, Tamrat Endale, Yolanda R. Smith.

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
