## [Decision Letter · Decision Letter 0]

30 Jan 2023

PONE-D-22-34373Scoping review of qualitative studies investigating reproductive health knowledge, attitudes, and practices among men and women across RwandaPLOS ONE

Dear Dr. Buser,

Thank you for submitting your manuscript to PLOS ONE. After careful consideration, we feel that it has merit but does not fully meet PLOS ONE’s publication criteria as it currently stands. Therefore, we invite you to submit a revised version of the manuscript that addresses the points raised during the review process.

We look forward to receiving your revised manuscript.

Kind regards,

Ashfikur Rahman

Academic Editor

PLOS ONE

Journal Requirements:

-https://www.unfpa.org/sexual-reproductive-health?page=3&type_1=All

-https://fp2030.org/sites/default/files/Rwanda%20FP-ASRH_Strategic%20Plan%202018%20-2014%20final.pdf

In your revision ensure you cite all your sources (including your own works), and quote or rephrase any duplicated text outside the methods section. Further consideration is dependent on these concerns being addressed.

Reviewers' comments:

Reviewer's Responses to Questions

**Comments to the Author**

1. Is the manuscript technically sound, and do the data support the conclusions?

Reviewer #1: No

2. Has the statistical analysis been performed appropriately and rigorously? 

Reviewer #1: N/A

3. Have the authors made all data underlying the findings in their manuscript fully available?

Reviewer #1: Yes

4. Is the manuscript presented in an intelligible fashion and written in standard English?

Reviewer #1: No

5. Review Comments to the Author

Reviewer #1: Reviewer’s Comments

Thank you very much for giving me a chance to review the manuscript entitled “Scoping review of qualitative studies investigating reproductive health knowledge, attitudes, and practices among men and women across Rwanda” to the journal of “PLOS ONE”. I appreciate the time and effort that the authors have dedicated to preparing the manuscript. I read the manuscript very carefully and consequently raised the quarries and suggestions to improve the manuscript which are as follows:

Major Revisions:

1. In ‘Introduction’ a clear explanation is needed about the background of the study and the importance of this study (Line no.124).

2. The heading of methods should be materials and methods according the guideline of PLOS ONE Journal (Line no. 157).

3. In the ‘method’ section, the authors have mentioned that the search was limited to articles published from 2002 to 2022 (Line no. 175-176). So the authors should provide the logic of limiting the search within this timeframe.

4. Under the sub-heading of ‘Screening Process’, the authors mentioned that the title and abstract screening was performed by two independent researchers (Line no. 194-195). So they have to specify who actually performed the title and abstract screening process. Similarly, the authors have to clearly specify who actually reviewed the full-text articles instead of using and/or vague term like (JMB, EA, OT, FGD, RG, GK, LI, TE, 197 and/or YS) (Line no. 196-197).

5. The authors are suggested to include the ‘Study quality assessment/Quality assessment of the included literature’ point under ‘materials and methods’ section (Line no. 157). They can follow the critical appraisal skill program (CASP) checklists (CASP, 2018) to assess the quality of the qualitative studies.

6. The ‘result’ section has to revise and extend in line with the specific objectives. It would be better to add relevant quotations (if possible) from included studies under different themes (Line no. 204).

7. The ‘Discussion’ section needs major revision. Here, the authors can compare and contrast their study findings with previous studies more extensively (Line no. 312).

8. The authors have to revise the conclusion and have to specify the results-specific policy recommendations (Line no. 334).

9. Overall, the writing of the manuscript has to improve following Standard English and ‘PLOS ONE’ Journal’s guidelines.

The author(s) are suggested to rewrite the manuscript accordingly. Hopefully, the aforesaid comments and suggestions would help to enrich this manuscript.

6. PLOS authors have the option to publish the peer review history of their article (what does this mean?). If published, this will include your full peer review and any attached files.

Reviewer #1: No

<quillbot-extension-portal></quillbot-extension-portal>

---

## [Author Response · Author response to Decision Letter 0]

24 Feb 2023

23 February 2023

PLOS ONE Editorial Board

Title: "Scoping review of qualitative studies investigating reproductive health knowledge, attitudes, and practices among men and women across Rwanda” (PONE-D-22-34373)

Dear Md. Ashfikur Rahman: 

We are very pleased to have the opportunity to revise our original manuscript for publication in PLOS ONE. We addressed each of the reviewer’s comments and made the necessary revisions and modifications. Responses are provided below.

Journal Requirements:

Requirement #1: Please ensure that your manuscript meets PLOS ONE's style requirements, including those for file naming. The PLOS ONE style templates can be found at 

Response #1: Thank you, we ensured that our manuscript meets PLOS ONE's style requirements and updated the file names. 

Requirement #2: We noticed you have some minor occurrence of overlapping text with the following previous publication(s), which needs to be addressed:

-https://www.unfpa.org/sexual-reproductive-health?page=3&type_1=All

-https://fp2030.org/sites/default/files/Rwanda%20FP-ASRH_Strategic%20Plan%202018%20-2014%20final.pdf

In your revision ensure you cite all your sources (including your own works), and quote or rephrase any duplicated text outside the methods section. Further consideration is dependent on these concerns being addressed.

Response #2: Thank you, in our revision we revised the overlapping text for UNFPA and FP2030 citations. We also ensured that we cited all our sources outside the methods section. 

Requirement #3: Please include captions for your Supporting Information files at the end of your manuscript, and update any in-text citations to match accordingly. Please see our Supporting Information guidelines for more information: http://journals.plos.org/plosone/s/supporting-information. 

Response #3: We updated the text to include captions at the end of the manuscript for the Supporting Information files. 

Reviewer comments:

Reviewer #1

Comment #1: In ‘Introduction’ a clear explanation is needed about the background of the study and the importance of this study (Line no.124).

Response #1: Thank you for your comment. To improve the manuscript, a statement was added at the beginning of the introduction to summarize the need and importance of the study, which was also integrated throughout the introduction.

Comment #2: The heading of methods should be materials and methods according the guideline of PLOS ONE Journal (Line no. 157).

Response #2: Thank you for sharing this observation. We updated the heading of this section to adhere to submission guidelines. 

Comment #3: In the ‘method’ section, the authors have mentioned that the search was limited to articles published from 2002 to 2022 (Line no. 175-176). So the authors should provide the logic of limiting the search within this timeframe.

Response #3: We appreciate this suggestion for improvement and revised the text providing the logic of limiting the search within this timeframe.

Comment #4: Under the sub-heading of ‘Screening Process’, the authors mentioned that the title and abstract screening was performed by two independent researchers (Line no. 194-195). So they have to specify who actually performed the title and abstract screening process. Similarly, the authors have to clearly specify who actually reviewed the full-text articles instead of using and/or vague term like (JMB, EA, OT, FGD, RG, GK, LI, TE, 197 and/or YS) (Line no. 196-197).

Response #4: We updated the text to mention who performed the title and abstract screening process. We also removed the vague term and/or to clarify that all authors listed in brackets actually reviewed the articles. 

Comment #5: The authors are suggested to include the ‘Study quality assessment/Quality assessment of the included literature’ point under ‘materials and methods’ section (Line no. 157). They can follow the critical appraisal skill program (CASP) checklists (CASP, 2018) to assess the quality of the qualitative studies.

Response #5: We updated the text to incorporate this suggestion and indicated that we followed the CASP qualitative study checklist. 

Comment #6: The ‘result’ section has to revise and extend in line with the specific objectives. It would be better to add relevant quotations (if possible) from included studies under different themes (Line no. 204).

Response #6: We thank the reviewer for sharing this suggestion for improvement. We revised the ‘result’ section by adding relevant quotations throughout from included studies under different themes.

Comment #7: The ‘Discussion’ section needs major revision. Here, the authors can compare and contrast their study findings with previous studies more extensively (Line no. 312).

Response #7: We expanded the discussion to compare our study findings with previous studies more extensively.

Comment #8: The authors have to revise the conclusion and have to specify the results-specific policy recommendations (Line no. 334).

Response #8: Thank you, we revised the conclusion and specified the results-specific policy changes recommended. 

Comment #9: Overall, the writing of the manuscript has to improve following Standard English and ‘PLOS ONE’ Journal’s guidelines.

Response #9: Thank you, we revised the text throughout to improve English language usage and ‘PLOS ONE’ submission guidelines.

Sincerely,

The Authors

---

## [Editor Report · Decision Letter 1]

20 Mar 2023

Scoping review of qualitative studies investigating reproductive health knowledge, attitudes, and practices among men and women across Rwanda

PONE-D-22-34373R1

Dear Dr. Buser,

We’re pleased to inform you that your manuscript has been judged scientifically suitable for publication and will be formally accepted for publication once it meets all outstanding technical requirements.

Kind regards,

Ashfikur Rahman

Academic Editor

PLOS ONE

Additional Editor Comments (optional):

Thanks for addressing the comments made by the reviewers, I have gone through the revision, and I think it is suitable for publication. I communicated w
---

## [Editor Report · Acceptance letter]

23 Mar 2023

PONE-D-22-34373R1 

Scoping review of qualitative studies investigating reproductive health knowledge, attitudes, and practices among men and women across Rwanda 

Dear Dr. Buser:

I'm pleased to inform you that your manuscript has been deemed suitable for publication in PLOS ONE. Congratulations! Your manuscript is now with our production department. 

Kind regards, 

on behalf of

Dr. Ashfikur Rahman 

Academic Editor

PLOS ONE